https://doi.org/10.1038/s42003-020-01320-6　**OPEN**

# Reward uncertainty asymmetrically affects information transmission within the monkey fronto-parietal network

Bahareh Taghizadeh[1,2,7], Nicholas C. Foley[3,4,7], Saeed Karimimehr[1,2], Michael Cohanpour[3,4], Mulugeta Semework[3,4], Sameer A. Sheth[5], Reza Lashgari[1,4] & Jacqueline Gottlieb [3,4,6✉]

A central hypothesis in research on executive function is that controlled information processing is costly and is allocated according to the behavioral benefits it brings. However, while computational theories predict that the benefits of new information depend on prior uncertainty, the cellular effects of uncertainty on the executive network are incompletely understood. Using simultaneous recordings in monkeys, we describe several mechanisms by which the fronto-parietal network reacts to uncertainty. We show that the variance of expected rewards, independently of the value of the rewards, was encoded in single neuron and population spiking activity and local field potential (LFP) oscillations, and, importantly, asymmetrically affected fronto-parietal information transmission (measured through the coherence between spikes and LFPs). Higher uncertainty selectively enhanced information transmission from the parietal to the frontal lobe and suppressed it in the opposite direction, consistent with Bayesian principles that prioritize sensory information according to a decision maker's prior uncertainty.

[1] Brain Engineering Research Center, Institute for Research in Fundamental Sciences, Tehran, Iran. [2] School of Cognitive Sciences, Institute for Research in Fundamental Sciences, Tehran, Iran. [3] Department of Neuroscience, Columbia University, New York, NY, USA. [4] Zuckerman Mind Brain Behavior Institute, Columbia University, New York, NY, USA. [5] Department of Neurosurgery, Baylor College of Medicine, Houston, TX, USA. [6] The Kavli Institute for Brain Science, Columbia University, New York, NY, USA. [7] These authors contributed equally: Bahareh Taghizadeh, Nicholas C. Foley. ✉email: jg2141@columbia.edu

Executive control is broadly understood as the ability to engage in information processing in pursuit of a goal, especially in circumstances requiring non-habitual or novel responses[1]. In humans and monkeys, executive function depends on a network of frontal and parietal areas, which is activated in relation to demanding behaviors requiring the suppression of inappropriate response tendencies, monitoring and adjusting behavioral strategies, and the goal-directed control of attention[1,2].

Theories of computational rationality, like current frameworks of executive function, propose that controlled (rather than automatic) information processing is costly and is engaged in proportion to the benefits it brings to the organism[1,3]. Because the decision-theoretic (Bayesian) definition of information is in terms of a reduction of uncertainty, an important implication of this view is that control should be optimally allocated to tasks that not merely have reward value but, more specifically, have uncertainty. It is in conditions of higher ex ante uncertainty that animals can expect to obtain the greatest benefits from processing new information and improving prediction accuracy[4–7].

Consistent with this view, a growing literature shows that attention is recruited by uncertainty independently of reward gains. Animals are intrinsically motivated to resolve uncertainty independently of instrumental incentives[5,8], the expectation of new information influences eye movements in humans[9] and monkeys[10], and oculomotor neurons in monkey parietal cortex have stronger responses preceding saccades that are expected to reduce uncertainty[11]. And yet, while existing studies have tested neural activity in the fronto-parietal network in tasks involving risk and ambiguity, learning, exploration, novelty, or surprise (e.g. refs. [12–17]), critical open questions remain about the cellular effects of uncertainty on this network.

One question concerns the distinction between uncertainty and reward gains. In instrumental conditions, when animals make reward-maximizing decisions, reductions of decision uncertainty are closely related with increases in long-term reward gains[5,18]. A handful of studies recently used non-instrumental conditions to show that individual neurons have distinct responses to the variance and value of expected rewards, but these studies have targeted the orbitofrontal cortex[19] and subcortical structures[20,21]

rather than the fronto-parietal network[22–26] (but see[11] for a notable exception).

A second key question is how uncertainty affects not only neural activity within areas but information flow between areas. A central tenet of Bayesian[4] and predictive coding theories[6] is that, in states of high prior uncertainty, the brain downregulates top-down signals conveying uncertain prior expectations and upregulates the bottom-up transmission of sensory information. However, while this view is prevalent in computational theories, there has been no empirical demonstration of uncertainty-dependent modulations of functional connectivity.

To examine these questions, we simultaneously recorded single-neuron responses and local field potential (LFP) oscillations in the dorsolateral prefrontal cortex (dlPFC) and area 7A, two strongly interconnected nodes of the monkey fronto-parietal network. We used a simple task in which monkeys were cued to expect certain or uncertain rewards but could not make decisions to maximize those rewards. We show that uncertainty has representations in action potential activity and LFP oscillations that are distinct from those of EV. Importantly, uncertainty asymmetrically enhances spike-field coherence (SFC) from the parietal to the frontal lobe while suppressing SFC in the opposite direction, consistent with theoretical predictions of optimal inference under uncertainty.

## Results

**Task and behavior**. Two monkeys performed a visually guided saccade task in which they formed expectations about the trial's rewards based on familiar visual cues. On each trial after achieving central fixation, the monkeys were shown a cue indicating the trial's reward probability (Fig. 1a), which was followed, after a 400 ms delay period, by presentation of the target for the subsequent saccade. Upon making the required saccade, the monkeys received a reward according to the probability signaled by the cue. Cue and target locations were independently randomized across two locations (8° eccentricity to the right or left of fixation). Thus, the cue was only predictive of the coming reward but not the instrumental action, allowing us to examine behavioral and neural responses to reward expectations independently of saccade planning or reward-maximizing decision strategies.

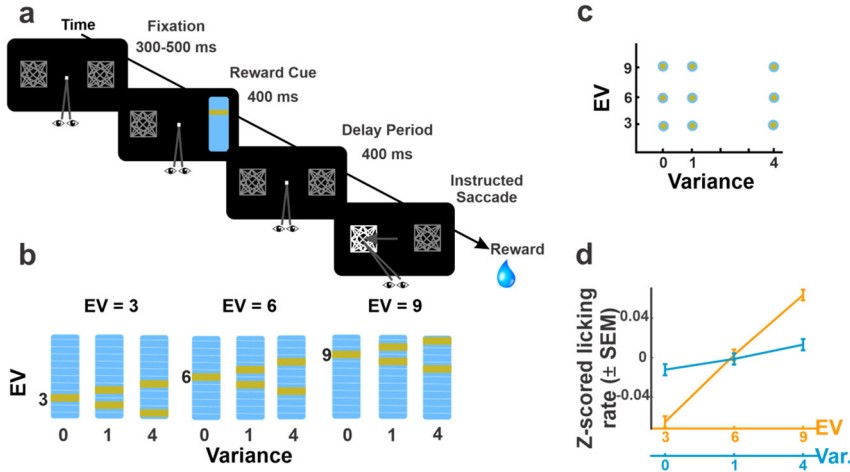

**Fig. 1 Task design and behavior. a** *Task*: The monkeys initiated a trial by looking at a fixation point flanked by two placeholders. A randomly selected placeholder was then replaced by a reward cue for 400 ms, followed by a 400-ms delay (memory) period, and presentation of the saccade target (luminance increase in a randomly selected placeholder). After making the instructed saccade, the monkeys received a reward according to the cued distribution. **b** *Visual cues*: Monkeys were familiarized with nine visual cues signaling the possible reward magnitudes on an 11-point scale (1 point = 0.1 mL of water). **c** *Orthogonalization of variance and EV*: Across the cue set, variance and EV could each take three discrete levels and were statistically dissociated. **d** *Anticipatory licking* before reward delivery increased as a function of both variance and EV. Points show average and SEM of the z-scored licking rates (n = 12,029 independent trials).

Monkeys were familiarized with a set of cues signaling nine reward distributions, whose variance and EV were statistically dissociated. Three cues signaled deterministic rewards of, respectively, 3, 6, or 9 points, whereas the remaining six cues indicated probabilistic rewards, with a small or large reward size being equally likely to occur (Fig. 1b single- and double-line cues). Reward sizes were determined through a mean-preserving procedure, whereby the small and large probabilistic rewards were symmetrically positioned with low or high variance, around the deterministic EV. This produced a cue set that statistically dissociated three levels of variance (0, 1, and 4) and three levels of EV (Fig. 1c). Both monkeys achieved high proficiency, with a high fraction of correctly completed trials (monkey 1: 83% correct, monkey 2: 81% correct overall). Importantly the fraction of correct trials did not vary with variance or EV (two-way ANOVA for each monkey, all $p > 0.19$) ensuring that the rewards that the monkeys experienced were not distorted by uneven performance and corresponded to those signaled by the cues.

Analyses of anticipatory licking confirmed that the monkeys were familiar with the cues and were cognizant of both variance and EV (Fig. 1d). The generalized linear model (GLM) coefficients for licking (see "Methods") were significantly greater than zero for both variance and EV (mean ± SEM, variance: 0.03 ± 0.0006; EV: 0.24 ± 0.002, all $p < 10^{-9}$ relative to 0, signed-rank test). In contrast, EV and variance did not consistently affect the monkeys' saccades. Saccade reaction times (RTs) increased with EV in monkey 1 but not in monkey 2 (GLM coefficients, respectively, $p = 2.6 \times 10^{-8}$ and $p = 0.35$ relative to 0) and were not affected by variance in either monkey (all $p > 0.1$). Moreover, the effects of EV and variance on licking were uncorrelated with those on RT across sessions and showed no significant interactions with the location of the visual cue (all $p > 0.3$). Thus, reward expectations affected anticipatory licking independently of saccade orienting, consistent with previous reports that the two behaviors have different reward sensitivity[27].

To investigate the neural correlates of variance and EV, we implanted multi-channel electrode arrays in area 7A and the dlPFC focusing on subdivisions that are reciprocally connected and have visual and attention-related activity—i.e., area OPT in the parietal cortex and the pre-arcuate portion of the dlPFC[28,29] (Fig. S1). We describe the effects of variance and EV in single-neuron activity, followed by their influence on LFP oscillatory power and SFC.

**Variance and EV have distinct single-neuron representations**. In both 7A and dlPFC, individual neurons showed significant encoding of variance or EV (Table 1 and Fig. 2). The sensitive neurons were equally likely to respond to uncertainty and EV with increases or decreases in firing (Table 1 and Fig. 2). Importantly, the GLM coefficients capturing each effect (see "Methods") were uncorrelated, suggesting that variance and EV are encoded in distinct populations of cells (7A: $r = 0.03$, $p = 0.49$, $n = 522$; dlPFC: $r = 0.06$, $p = 0.15$, $n = 530$).

Responses with positive and negative scaling had similar prevalence and strength in the two areas (Fig. 2 and Table 1) and, across all the cells, the average coefficients showed no net enhancement or suppression of firing with either variable (all $p > 0.23$, with the single exception of a net positive effect of EV in area 7A; mean ± SEM GLM coefficient of 0.171 ± 0.032, $p = 10^{-6}$ relative to 0). Cells with positive and negative scaling had sustained effects throughout the cue and delay epochs (Fig. 3a). We found no correlation between trial-by trial firing rates and licking responses, suggesting that the cells encoded expectations rather than licking per se. Moreover, cells with positive or negative scaling for one variable had no significant sensitivity to

the other factor (Fig. 3b), confirming that variance and EV were encoded by distinct populations of cells.

Because cue location was included as a nuisance regressor in the GLM, the EV, and variance sensitivity were above and beyond any cue location response. Four additional observations support this conclusion. First, while some neurons encoded the location of the visual cue, location coefficients were uncorrelated with those for variance or EV (Fig. S2). Second, the EV and variance selective cells showed no consistent visual response, ruling out that they merely encoded the appearance of the cue (Fig. S2). Third, EV and variance coefficients that were estimated separately for each cue location were statistically equivalent (all $p > 0.59$ sign-rank test) and highly correlated (all $p < 0.02$). Finally, we found no significant correlation between trial-by-trial firing rates and saccadic RT. Thus, the neurons encoded global expectations of reward variance and EV independently of visuo-spatial selectivity or saccade planning activity.

*Noise correlations:* Given the stark segregation of EV and variance responses we found in both areas, we wondered whether the neurons encoding these variables had distinct functional connectivity. To examine this question, we computed noise correlations between trial-by-trial activity in pairs of simultaneously recorded cells, focusing on firing rates in a 600 ms pre-cue epoch preceding cue onset to avoid confounds related to evoked activity[30].

Noise correlations were higher in pairs in which both neurons coded for the same factor (both neurons encoding variance or both encoding EV) relative to pairs with mixed selectivity (Fig. 4a, "across-factor" vs "within factor") and this difference was highly robust in both areas (Table 2, dlPFC, $p = 2.5 \times 10^{-7}$, $n = 47$ and $n = 56$ pairs; 7A, $p = 5.1 \times 10^{-8}$, $n = 79$ and $n = 43$ pairs; Kruskal–Wallis test). In addition, in pairs with homogeneous selectivity, noise correlations were larger if the two neurons had the same versus opposite polarity (Fig. 4a) for both variables and both areas (Table 2). Variance, EV, or response polarity had no effect on across-trial variability (Fano factor), ruling out that this may have produced apparent effect on noise correlations. Thus, subject to their encoding polarity, neurons responding to variance shared distinct variability relative to those encoding EV.

*Decoding*: Because information can be transmitted by neurons that lack linear selectivity, we conducted a final analysis to estimate the decoding capacity from the entire population of cells. We trained support vector machine (SVM) classifiers to perform pairwise discriminations between the different levels of variance and EV based on the population responses and analyzed the boostrapped distributions of excess accuracy (the differences in accuracy in the real and label-shuffled (null) data sets). To determine the extent to which variance and EV had distinct or overlapping representations, we also tested incongruent train-testing regimes—training the classifiers on variance and EV and testing on the untrained variable.

Decoding performance in congruent training-testing regimes was clearly superior to that in incongruent regimes for both variables in both areas. In both the pooled analyses (Fig. 4b) and pairwise comparisons (Fig. S3), 95% confidence bands were clearly above 0 for all congruent train-testing classifications, while decoding in incongruent regimes was significantly weaker and at chance levels in all cases. There were no significant differences between the decoding of variance and EV in 7A and dlPFC.

In sum, analysis of single-neuron activity, noise correlation, and population decoding show that variance and EV had clearly segregated representations that were similar in 7A and the dlPFC.

**Variance and EV modulate oscillatory LFP power**. Because, in addition to spiking activity, oscillatory LFP potentials are

**Table 1 Single-neuron sensitivity to variance and EV in dlPFC and 7A.**

| | Proportion significant | | | $\beta$-values (mean ± SEM) | | |
|---|---|---|---|---|---|---|
| | dlPFC | 7A | dlPFC vs 7A | dlPFC | 7A | dlPFC vs 7A |
| **EV** | | | | | | |
| EV+ | 5.7% (30) | 12% (62) | $\chi^2 = 2.410$, d.f. $= 1$, $p > 0.1$ | 1.13 ± 0.13* | 1.32 ± 0.11* | $p = 0.72$ |
| EV− | 5.7% (30) | 6% (31) | $\chi^2 = 0.007$, d.f. $= 1$, $p > 0.9$ | −0.94 ± 0.07* | −0.80 ± 0.07* | $p = 0.06$ |
| All EV sensitive | 11.3% (60) | 18% (93) | $\chi^2 = 1.690$, d.f. $= 1$, $p > 0.1$ | 0.10 ± 0.15 | 0.61 ± 0.13* | $p = 0.70$ |
| **Risk** | | | | | | |
| Variance+ | 2.8% (15) | 2.3% (12) | $\chi^2 = 0.060$, d.f. $= 1$, $p > 0.1$ | 0.73 ± 0.09* | 0.79 ± 0.1* | $p = 0.62$ |
| Variance− | 3.4% (18) | 3.1% (16) | $\chi^2 = 0.017$, d.f. $= 1$, $p > 0.9$ | −0.72 ± 0.08* | −0.95 ± 0.1* | $p = 0.10$ |
| All var. sensitive | 6.2% (33) | 5.4% (28) | $\chi^2 = 0.070$, d.f. $= 1$, $p > 0.1$ | −0.06 ± 0.14 | −0.20 ± 0.18 | $p = 0.08$ |
| **Interaction** | | | | | | |
| All interaction | 6.6% (35) | 6.1% (32) | $\chi^2 = 0.018$, d.f. $= 1$, $p > 0.9$ | 0.04 ± 0.18 | 0.82 ± 0.23* | $p = 0.04$ |
| All modulations | 22% (117) | 26% (134) | $\chi^2 = 3.460$, d.f. $= 7$, $p > 0.1$ | | | |

The left half of the table ("proportion significant") shows the percentage (number) of neurons with significant coefficients, and the results of chi-square tests of proportions comparing 7A and dlPFC. The right half ("$\beta$-values") shows the mean ± SEM of the signed coefficients in the sensitive cells and the results of Kruskall–Wallis non-parametric analysis of variance comparing the two areas. Note that, while the individual area averages refer to the *signed* values of the coefficients, the comparison between areas is on the *absolute* values to indicate whether the effects are stronger in any one area regardless of sign.

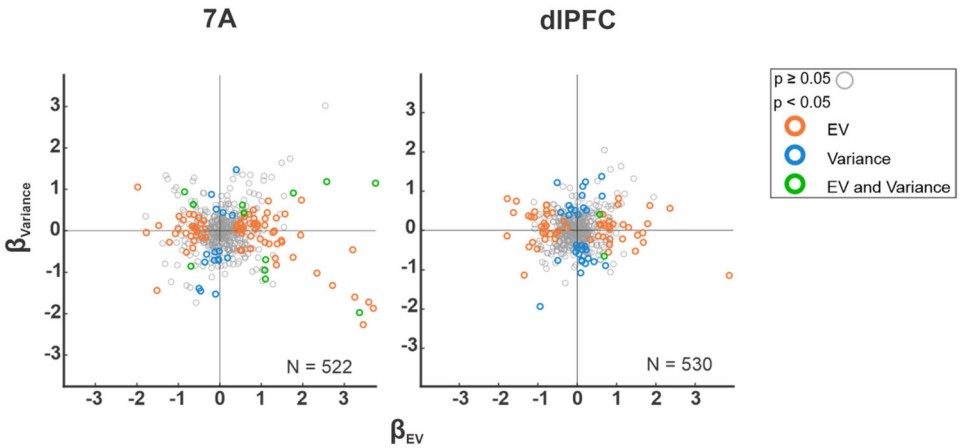

**Fig. 2 Single-neuron encoding of variance and EV is uncorrelated.** Scatterplots of the GLM coefficients capturing the effect of EV ($\beta_{EV}$, abscissa) and variance ($\beta_{Variance}$, ordinate) in 7A and dlPFC. Every point is one neuron, with the number of observations indicated in each plot. The colors indicate the significance of individual EV and Var coefficients as noted by the legend.

sensitive indicators of cognitive states[31–33] we next examined how oscillations are affected by variance and EV. To this end, we divided single-trial LFP traces into 1 Hz × 1 ms pixels spanning the cue and delay epochs and, for each pixel, fit a GLM model that included variance and EV as factors, controlling for cue location and interactions (identical to the model applied to spiking activity; "Methods"). The resulting coefficient maps showed that variance and EV exerted consistent effects in two frequency bands: a lower frequency band between 8 and 18 Hz, corresponding to α/low-β frequencies, and a higher band of 18–43 Hz, corresponding to the high-β/low-γ frequencies (Figs. 5 and 6).

Power in an α/low-β frequency band (8–18 Hz) is widely associated with task engagement and arousal in different tasks and brain areas in humans and monkeys[34]. Consistent with this widely replicated result, activity in this band was suppressed by variance and EV in both 7A and the dlPFC (Fig. 5, pink regions of interest (ROIs)). The strongest effects arose in the late cue and early delay periods (Fig. 6a, b) and were highly significant for both variables for each monkey (7A variance: monkey 1: $p < 6 \times 10^{-6}$ (Wilcoxon rank-sum test relative to 0 across all pixels in the ROI); monkey 2: $p < 2 \times 10^{-14}$; EV: monkey 1: $p < 2 \times 10^{-21}$, monkey 2: $p < 9 \times 10^{-13}$; dlPFC variance: monkey 1: $p < 6 \times$

$10^{-10}$, monkey 2: $p < 8 \times 10^{-8}$; EV: monkey 1: $p < 5 \times 10^{-28}$, monkey 2: $p < 2 \times 10^{-12}$).

In contrast with the uniform suppression in the low-frequency band, the effects in the high-β/low-γ differed for variance and EV and across the two areas (Fig. 6c vs d and Fig. 5, purple ROI). In 7A, power in this band was suppressed by variance and enhanced by EV (Fig. 6c vs d, dashed traces), while the dlPFC showed the opposite pattern—being enhanced by variance and suppressed by EV (Fig. 6c vs d, solid traces). Each effect was highly robust in each monkey (7A variance: monkey 1: $p < 2 \times 10^{-13}$, monkey 2: $p < 4 \times 10^{-17}$; 7A EV: monkey 1: $p < 2 \times 10^{-16}$, monkey 2: $p < 4 \times 10^{-18}$; dlPFC variance monkey 1: $p < 3 \times 10^{-6}$, monkey 2: $p < 2 \times 10^{-22}$; dlPFC EV: monkey 1: $p < 4 \times 10^{-28}$, monkey 2: $p < 9.5 \times 10^{-4}$). As for the single-neuron results, these effects were above and beyond location selectivity, were equivalent at the two cue locations and were uncorrelated with the sensitivity to variance and EV in saccadic RT. Thus, variance and EV reduced power in the α/low-β frequency range in both areas but had distinct area-specific effects in the high-β/low-γ frequency range.

**Variance enhances parietal-to-frontal information transmission.** Given theoretical predictions that uncertainty modulates the

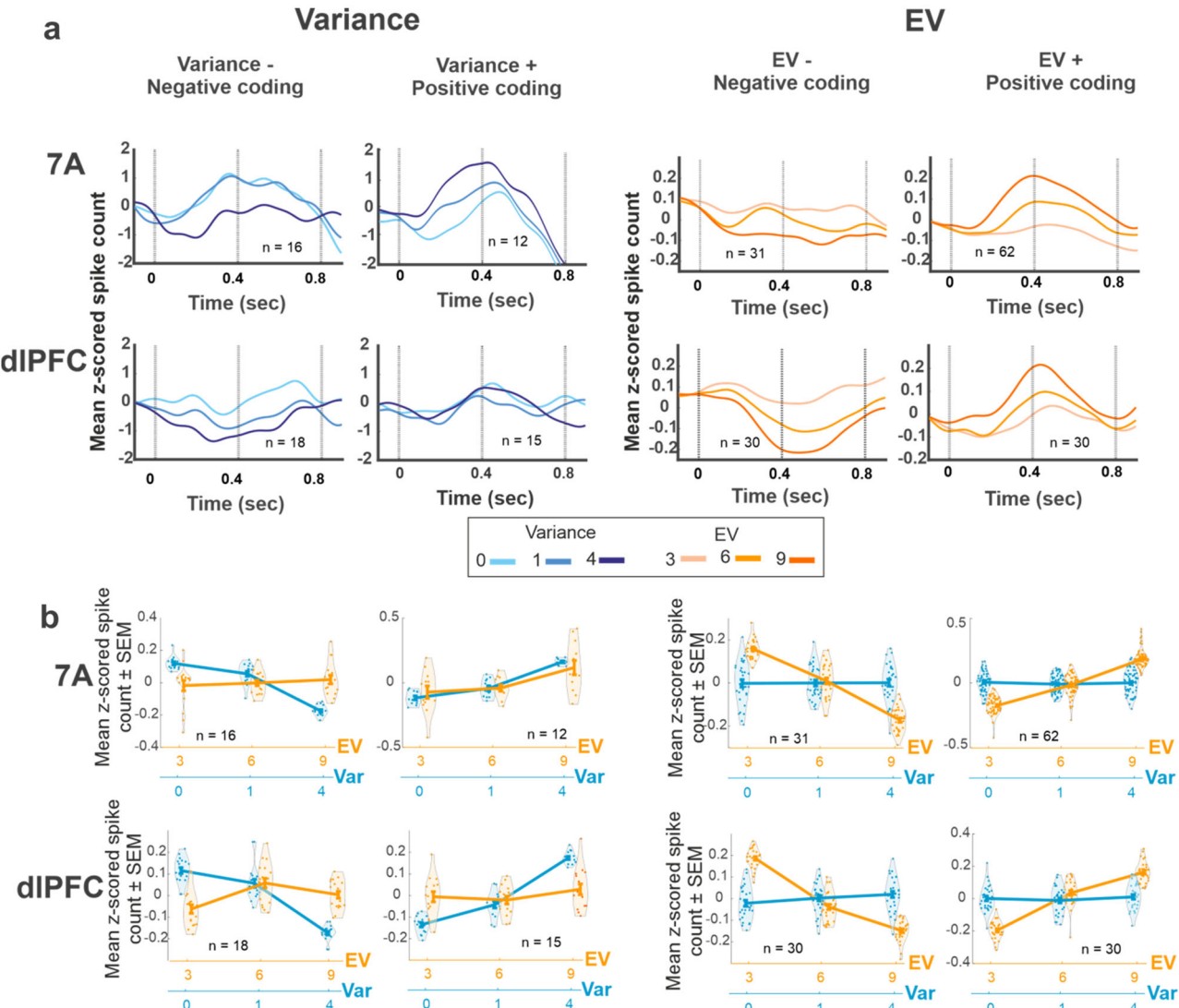

**Fig. 3 Single neurons had positive and negative encoding of variance and EV. a** Variance and EV encoding was sustained in the late cue and delay periods. Spike density functions for the subsets of cells with significant positive or negative coefficients for variance and EV (including the few cells with both main effects). "*n*" indicates the number of cells contributing to each trace. **b** *Spike counts* of the cells contributing to the traces in **a** grouped by variance (blue) and EV (orange). In the violin plots, each point shows the mean spike count of one cell in the interval of 0–800 ms after cue onset, *z*-scored relative to all the trials of that cell and averaged across trials with the indicated variance or EV. The thicker lines show the mean and SEM across cells.

balance between top-down and bottom-up information transmission[4,6], we asked how variance and EV modulate functional interactions among the two areas. To this end, we calculated SFC using the method of Vinck et al. that is known to compensate for biases due to low spike counts and volume conduction[35,36]. The SFC measures the extent to which spikes arrive at a consistent phase of the LFP oscillations and provides an index of directional interactions. The SFC between spikes in area A and LFPs in area B measures the extent to which outputs from area A influence area B, while the SFC between spikes in area B and LFPs in area A measure the opposite interactions[35–37] (see also "Methods").

The most robust modulation we found was an asymmetric effect of variance on fronto-parietal SFC. Higher uncertainty was associated with enhanced SFC from 7A spikes to dlPFC LFPs, suggesting enhanced information transmission from 7A to the dlPFC (Fig. 7a). Conversely, higher variance was associated with reduced SFC in the opposite direction, suggesting reduced information transmission from dlPFC to 7A (Fig. 7b). These effects were consistent in both monkeys and

could not be explained by changes in LFP power, which had opposite signs in the two areas (Figs. 5 and 6) or by LFP–LFP coherence, which did not show consistent modulations with variance or EV (Fig. S5). The SFC modulations were unique to variance and to across-area communications, with only weak and inconsistent effects being produced by EV on SFC across areas (Fig. S4a) and by both variance and EV within areas (Fig. S4b).

The SFC modulations by variance extended to all frequency bands and differed across the task epochs (Fig. 7c–f). In the α/low-β frequency band, the earliest modulation was an increase in parietal-to-frontal SFC followed by a decrease in the frontal-to-parietal direction (Fig. 7c, f; all $p < 10^{-7}$ in each monkey, Krusal–Wallis test; see figure legend for detailed statistics). In the high-β/low-γ frequency band this sequence was reversed, with the earliest modulation being reduction in frontal-to-parietal SFC followed by increased parietal-to-frontal SFC (Fig. 7 d, e; all $p < 10^{-7}$ in each monkey). Thus, uncertainty sets off an intricate temporal sequence of increases and decreases in fronto-parietal functional connectivity.

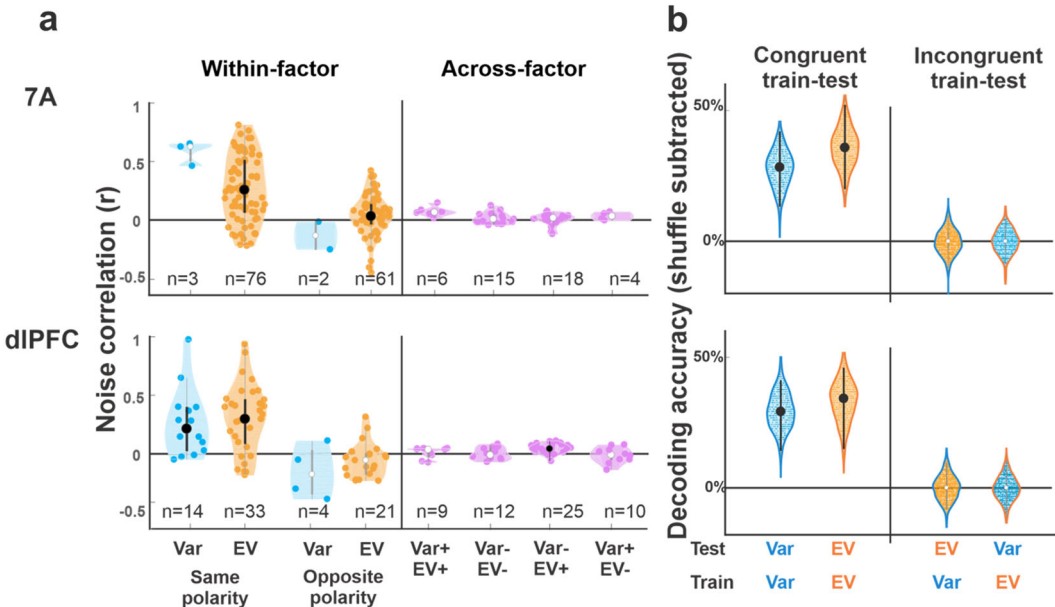

**Fig. 4 Noise correlations and population decoding support separate representations of variance and EV. a** *Noise correlations:* Each violin plot shows the correlation coefficient for pairs of cells that were simultaneously recorded and had specific combinations of selectivity as noted on the x-axis. The numbers show the number of pairs in each distribution. The dots in each distribution show individual pairs; the larger points and whiskers show the median coefficient and 25th and 75th percentiles. Distributions that are significantly higher than 0 ($p < 0.05$) are shown with black dots and whiskers, otherwise they are shown with white dots and gray whiskers. "Within-factor" distributions show pairs in which both cells were selective to EV or both were selective for variance, further separated by whether the two cells had the same encoding polarity (EV+/EV+ or EV−/EV−, variance+/variance+ or variance −/variance−) or opposite polarity (EV+/EV− or variance+/variance−). "Across-factor" distributions show the coefficients for pairs in which one cell was selective for EV and the other for variance, further separated by polarity as noted. **b** *Classification accuracy based on population responses:* Each violin plot shows the distribution of accuracy across 200 bootstrap iterations (after subtracting the accuracy in a randomized dataset). The large dot and error bars show the average accuracy and 95% confidence intervals, with above-chance classification shown with black dots and whiskers. The different distributions correspond to different train/test regimes, as indicated by the x-axis and colors (test variable: dot color; train variable: outline color; orange: EV; blue: variance).

**Table 2 Noise correlations.**

|  | r values (mean ± SEM) | | |
|---|---|---|---|
|  | **dlPFC** | **7A** | **dlPFC vs 7A** |
| EV/EV |  |  |  |
| Same polarity | **0.30 ± 0.05\*\*, N = 33** | **0.27 ± 0.03\*\*, N = 76** | $p = 0.66$ |
| Opposite polarity | −0.05 ± 0.03, N = 21 | 0.04 ± 0.02\*, N = 61 | $p = 0.007$ |
| Same vs opp. (p value) | **$p = 1.8 \times 10^{-5}$** | **$p = 2.56 \times 10^{-6}$** |  |
| Var/Var |  |  |  |
| Same polarity | **0.26 ± 0.08a\*\*, N = 14** | 0.58 ± 0.06, N = 3 | $p = 0.04$ |
| Opposite polarity | −0.16 ± 0.11, N = 4 | −0.14 ± 0.11, N = 2 | $p = 0.64$ |
| Same vs opp. (p value) | $p = 0.015$ | $p = 0.083$ |  |
| EV/Var |  |  |  |
| Same polarity | −0.0006 ± 0.01, N = 21 | 0.02 ± 0.01, N = 21 | $p = 0.20$ |
| Opposite polarity | 0.02 ± 0.01\*, N = 35 | 8e−4 ± 0.01, N = 22 | $p = 0.07$ |
| All EV/Var vs all [EV/EV and Var/Var same polarity] | **$P = 2.5 \times 10^{-7}$** | **$p = 5.1 \times 10^{-8}$** |  |

Each entry shows the average and SEM of the Pearson correlation coefficient, and the number (N) of simultaneously recorded cell pairs of each type that met the analysis criteria (see "Methods"). Stars and bold typeface indicate the results of signed-rank tests relative to 0.
\*\*$p < 0.01$, \*$p < 0.05$.

**Individual variability and risk preference**. Although our study did not examine economic decisions, it is interesting to consider how the responses to reward expectancy we report may relate to risk preference. To explore this question we tested the monkeys, after neural recordings were complete, on a choice version of the task in which the monkeys received two cues on each trial and chose one cue whose reward probability they wished to obtain (Fig. S6 legend). This revealed that monkey 1 was risk seeking and monkey 2 was risk averse—a highly significant individual

difference (% of choices to the higher variance of, respectively, 56.2% and 47.8%; both $p < 0.014$ signed-rank test against 50%; $p < 10^{-9}$ between monkeys; Fig. S6).

These individual differences corresponded with the relative sensitivity to variance versus EV in several behavioral and neural indicators. Monkey 1, who was risk seeking, was relatively more sensitive to EV rather than variance in his licking and saccadic RT; monkey 2, who was risk averse, showed the opposite pattern, and was more sensitive to variance relative to EV (Fig. S6).

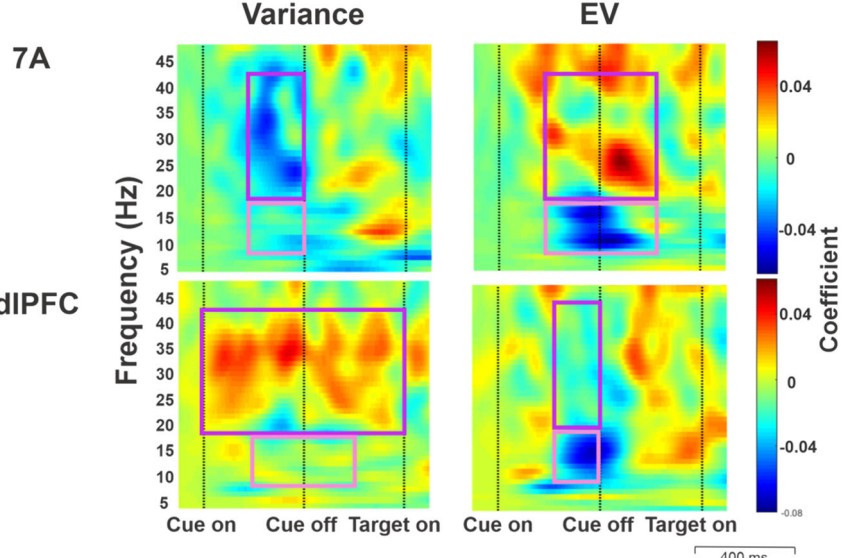

**Fig. 5 Variance and EV affect LFP oscillations in the α, β, and γ frequency bands.** Time-frequency maps of GLM coefficients indicating the effects of variance and EV on LFP power. Each pixel is a GLM coefficient obtained from fitting the LFP power spectrum across all the trials available in each monkey and averaging across monkeys (see "Methods, LFP pre-processing"). The rectangles superimposed on each map indicate the regions of interest (ROI) where consistent effects were found in both monkeys, in the α/low-β (8–18 Hz, pink) and high-β/low-γ (18–43 Hz, purple) frequency bands.

Among neural indicators, analogous differences were reflected in LFP power and the fraction of sensitive cells in both areas (Fig. S6) although not in measures of SFC. Thus, the relative weight that individuals afford to variance of EV may relate to risk attitudes—a conclusion that can be verified in future investigations.

## Discussion

We show that the uncertainty of an expected reward, independently of the value of the reward, affects multiple aspects of microscopic and mesoscopic fronto-parietal activity, including single-neuron responses, LFP oscillations, and SFC.

Uncertainty powerfully modulated LFP power, producing different effects in low- and high-frequency bands. Low-frequency-α/low-β-LFP power homogenously decreased in 7A and the dlPFC as a function of EV and uncertainty. Because lower α/low-β LFP power has been linked with enhanced task engagement, reduced inhibition, and desynchronized neural activity in multiple structures[38–40] this suggests that arousal was enhanced by both EV and uncertainty in our task. In contrast with the homogeneous effects of uncertainty in the lower frequency band, the signature of uncertainty in the higher frequency range differed markedly by area and was clearly distinct from that of EV. This heterogeneity is consistent with the diverse modulations previously reported for γ-band power, which consist of increases and decreases with attention across tasks and cortical areas[38]. Based on the prevailing view that γ-band oscillations primarily index feedforward sensory processing[32,41] our findings suggest that feedforward processing is differentially affected by uncertainty versus EV.

Another clear distinction we find is that variance and EV had separate representations in spiking activity. Previous studies have shown that reward variance is encoded independently of value by individual neurons in the orbitofrontal cortex[19], in subcortical structures such as the basal forebrain[20,21,42] and, more recently, in the anterior cingulate cortex[43,44]. Our findings show that this segregation extends to lateral fronto-parietal areas and to circuit-based measures including noise correlations and population decoding capacity.

Importantly, we show that, rather than producing overall increases or decreases in firing, uncertainty and EV had opponent-coding representations, enhancing or suppressing responses in distinct classes of cells. An opponent-coding representation has been previously reported for EV in the dlPFC[45,46] and here we show that it extends to the parietal cortex and to uncertainty. Our finding that neurons with similar polarity have higher noise correlations suggests that they form subnetworks with distinct functional properties. Neurons with positive and negative EV scaling may be associated with, respectively, approach and avoidance behaviors (go/no-go tendencies) that are mediated by distinct basal ganglia pathways[47]. Neurons with positive or negative variance coding, on the other hand, may arbitrate between different modes of cognitive control based on uncertainty—relying on a simpler striatal controller when familiar, habitual strategies are sufficient but engaging the prefrontal cortex in uncertain conditions[48].

While uncertainty and value have been shown to modulate the representations of specific objects or actions (e.g., in the frontal cortex[14,49] and dorsal striatum[50]) the EV and variance-sensitive cells we describe reflected global, non-spatial states of expectancy that were independent of visuo-spatial selectivity. This result most likely reflects the task we employed, in which monkeys merely formed expectations without formulating a choice, contrasting with previous studies in which monkeys made a deliberate choice. Thus, an important question for future investigations concerns the relation between expectancy and decision making, especially given our preliminary finding that the relative impact of uncertainty versus EV on expectations may be systematically related to individual risk attitudes.

A central result we report is that uncertainty had powerful effects on fronto-parietal connectivity. Higher uncertainty was associated with reduced SFC from the frontal to the parietal cortex but enhanced SFC from the parietal to the frontal lobe. These results are consistent with a recent report that, although fronto-parietal areas have similar single-neuron activity, the direction of their functional interactions can be strongly dependent on context[28]. That study found that, in monkeys performing a familiar categorization task, information about task context and

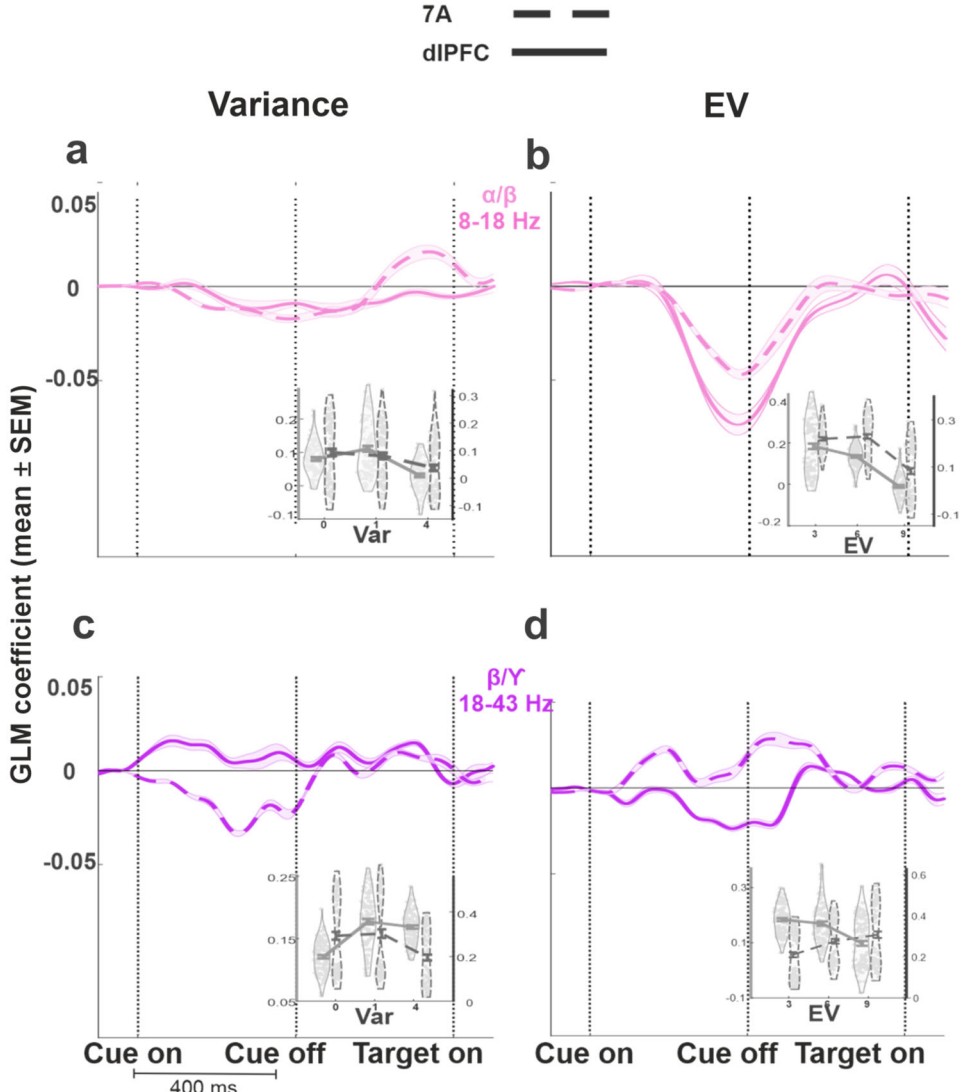

**Fig. 6 Variance and EV effects in low and high-frequency bands.** Each trace shows the average GLM coefficient in the α/low-β (8–18 Hz, pink, **a** and **b**) and high-β/low-γ (18–43 Hz, purple, **c** and **d**) (mean and SEM across the frequencies). The insets show the raw LFP power as a function of variance or EV in the ROIs shown in Fig. 5. The left gray axis and gray curve correspond to dlPFC, and, the right black axis and dashed black curve correspond to 7A. For these plots, z-transformed LFP power was first averaged within each electrode across all the trials collected on that electrode, and the data show average and SEM across n = 96 electrodes (48 in each monkey) as a function of the variable of interest (variance and EV). Violin plots show individual data points.

rules was predominantly transmitted in a top-down direction, from dlPFC to 7A. This is consistent with our finding that, when monkeys have low uncertainty, frontal-to-parietal SFC is higher than parietal-to-frontal SFC (i.e., 0 variance, Fig. 7a, b). However, we show that, for higher uncertainty, this balance can drastically change in favor of parietal-to-frontal transmission, consistent with Bayesian optimal inference theories.

Our finding that uncertainty reduces SFC in the top-down direction does not imply that the dlPFC goes "offline" in uncertain conditions. Indeed, the modulations of frontal-to-parietal and parietal-to-frontal SFC had overlapping time-courses and some of the strongest effects of uncertainty—i.e., on high-β/low-γ frequency LFP power (Fig. 5) and SFC (Fig. 7c–f)—emerged first in the dlPFC and only later in 7A. It is thus possible that the connectivity changes relied on dynamic interactions between the frontal and parietal lobes. We propose that uncertainty is detected by frontal cortical areas including the dlPFC and the anterior cingulate cortex; these areas may provide the initial drive which,

perhaps by triggering release of neuromodulators, ultimately leads to increases in sensory gains and enhancements in parietal-to-frontal transmission[2,51].

Our findings also support the idea that the parietal cortex plays an important role in resolving uncertainty. Early support for this view comes from the reinforcement learning literature showing that rats have increases in associability (learning rates) for uncertain sensory cues and these increases are reduced by lesions of the parietal cortex[52]. Subsequent single-neuron recordings in monkeys show that parietal neurons have enhanced responses to novel stimuli and salient distractors[16,53] and, in multi-step decision tasks, assign credit and learning specifically at junctures that resolve uncertainty[11,54,55]. Thus, an important direction for future research is to refine our understanding of the intricate mechanisms that allow the brain to allocate attention and other cognitive resources to task junctures that not only have high value but benefit from new information and a reduction of uncertainty[5,7,18,26].

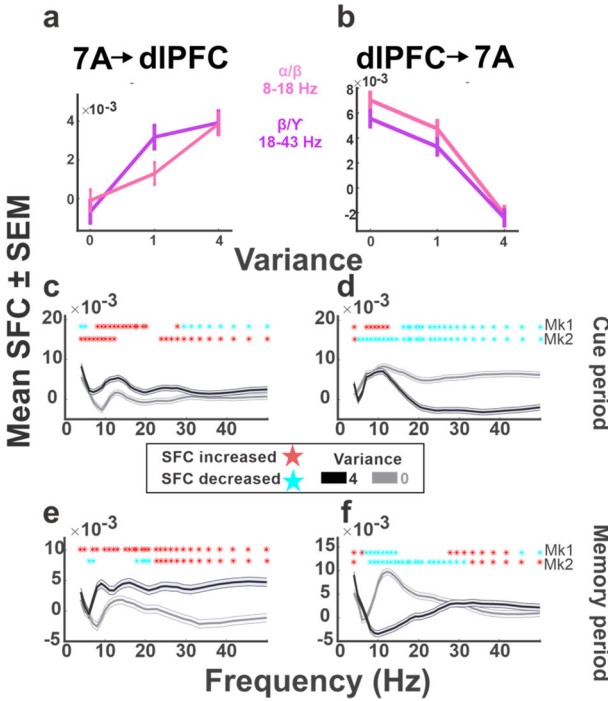

**Fig. 7 Uncertainty enhances spike-field coherence (SFC) from 7A to dlPFC but not vice versa. a** *SFC between 7A neurons and dlPFC LFPs as a function of variance*: Each point shows the mean and SEM of the SFC values in the time-frequency bins showing the most consistent effects of variance in the average data of both monkeys. **b** *SFC between dlPFC neurons and 7A LFPs in the same format as in* **a**. In **a** and **b**, SFC values were averaged across all neuron-LFP channel pairs and all task conditions with identical levels of uncertainty (*n* ranges between 136,368 and 164,880 independent pairs across conditions). **c–f** *SFC as a function of frequency for variance of 0 (gray) versus 4 (black)*. The curves show the mean and SEM of the SFC values for different frequencies in the time bins that showed the most consistent effects (**c**, **d** 200–400 ms; **f** 400–600 ms, and **e** 600–800 ms after cue on). The stars in each panel show the frequencies where SFC was significantly modulated for each monkey (Kruskal–Wallis test, $p < 0.05$ after correcting for multiple comparisons across frequencies). Blue stars indicate decrease and red stars indicate increase of SFC with variance. The SFC values for the peak effects in each panel are as follows (all units of $10^{-3}$). Cue period, mean ± SEM SFC for 7A → dlPFC in the α/low-β frequency band: $var_0$: $-0.09 \pm 0.054$, $var_4$: $3.8 \pm 0.5$; Kruskal–Wallis test, $p = 5.2 \times 10^{-8}$. For dlPFC→7A, in the high-β/low-γ frequency band: $var_0$: $5.5 \pm 0.6$, $var_4$: $-2.4 \pm 0.6$ ($p = 1.8 \times 10^{-21}$). Delay period, 7A → dlPFC in the high-β/low-γ frequency band: $var_0$: $-0.7 \pm 0.6$, $var_4$: $3.9 \pm 0.6$ ($p = 3.5 \times 10^{-7}$). dlPFC→7A, in the α/low-β frequency band: $var_0$: $7.0 \pm 0.6$, $var_4$: $-2.1 \pm 0.6$ ($p = 6.3 \times 10^{-16}$).

## Methods

**General methods**. Data were collected from two adult male rhesus monkeys (*Macaca mulatta*; 9–12 kg) using standard behavioral and neurophysiological techniques as described previously[56]. All methods were approved by the Animal Care and Use Committees of Columbia University and New York State Psychiatric Institute as complying with the guidelines within the Public Health Service Guide for the Care and Use of Laboratory Animals. Visual stimuli were presented on a MS3400V XGA high definition monitor (CTX International, Inc., City of Industry, CA; 62.5 × 46.5 cm viewing area). Eye position was recorded using an eye tracking system (Arrington Research, Scottsdale, AZ). Licking was recorded with an in-house device that detected interruptions in a laser beam produced by extensions of the monkeys' tongue.

**Task**. A trial started with the presentation of two textured square placeholders (1° width) located along the horizontal meridian at 8° eccentricity to the right and left of a central fixation point (white square, 0.2° diameter). After a 300–500 ms period of central fixation (when the monkeys maintained gaze within a 1.5–2° square window centered on the fixation point) one of the placeholders was replaced by a

randomly selected reward cue (a vertical rectangle measuring 1.2 × 5° with 11 gray bars indicating the reward scale, and one or two gradations highlighted in yellow, indicating the trial's rewards). The cue was visible for 400 ms and was followed by a 400-ms delay period, after which the fixation point disappeared and one of the placeholders simultaneously increased in luminance, indicating the saccade target. The target location was randomized independently of the reward cue. If the monkey made a saccade to the target with an RT of 100–700 ms and maintained fixation within a 2.0–3.5° window for 377 ms, he received a reward with the magnitude and probability that had been indicated by the cue.

**Neural recordings**. After completing behavioral training, each monkey was implanted with two 48-electrode Utah arrays (electrode length 1.5 mm) arranged in rectangular grids (1 mm spacing; monkey 1, 7 × 7 mm; monkey 2, 5 × 10 mm) and positioned in the pre-arcuate portion of the dlPFC and the posterior portion of area 7A (Fig. S1). Data were recorded using the Cereplex System (Blackrock, Salt Lake City, Utah) over 24 sessions spanning 4 months after array implantation in monkey 1, and 11 sessions spanning 2 months after implantation in monkey 2.

**Statistics and reproducibility**. Data were analyzed with MatLab (MathWorks, Natick, MA; version R2016-b) and other specialized software as noted below. Raw spikes were sorted offline using WaveSorter[57]. We analyzed a total of 12,029 trials that (1) were correctly completed and (2) had RT within 2 standard deviations relative to the mean of each monkey's full dataset (monkey 1: $n = 8082$ analyzed trials, monkey 2: $n = 3947$). These trials were further sub-selected for different analyses. For single-neuron analysis, we included only well-isolated cells, as defined by the automated sorting results followed by visual inspection to verify that only neurons with waveforms clearly separated from noise were included in the analysis, and that the population of cells was substantially different across days (ensuring that we did not systematically record from the same subsets of cells). For LFP and SFC analyses, trials were further cleaned to remove electrical artifacts as described in detail below. The unit of statistical comparison and statistical tests differed are described in detail throughout the text.

**Analysis of behavior**. The lickometer signal was digitized at 1 kHz to produce a trial-by-trial record of licking with 1 ms resolution. The probability of licking was measured in a time window centered on the time of each monkey's average peak licking response (monkey 1: 400–800 ms after cue onset; monkey 2: 800–1100 ms after cue onset). Licking probabilities in individual trials were pooled across sessions and subjected to a GLM analysis with EV and variance, including cue position and the EV × variance interaction as nuisance regressors (using a binomial distribution and logit link function and implemented in the fitglm function in the MATLAB statistics toolbox). Models that included a parametric uncertainty regressor outperformed those that included only a binary indicator of probabilistic versus deterministic cues and are presented throughout the paper.

**Single neurons spike analysis**. Raw spikes were sorted offline using WaveSorter and produced a total of 1175 neurons in dlPFC (749 in monkey 1) and 971 neurons in 7A (755 in monkey 1). We focused the analysis on the subset of units that were well isolated, had at least five trials in each condition, and fired at least five spikes on average within the time interval from 500 ms before to 1000 ms after cue onset, comprising 530 neurons in dlPFC (432 in monkey 1) and 522 neurons in 7A (481 in monkey 1).

To measure neuronal selectivity, we fit each neuron's trial-by-trial spike count in the interval 0–800 ms after cue onset using a GLM with factors EV, variance, EV × variance, and Cue location, using a normal distribution and identity link function. To estimate changes in firing rate variability, we computed the Fano factor—the ratio of across-trial variability to the mean firing rate. Although the Fano factor was lower during the cue/delay relative to the pre-cue epochs, we found no consistent changes as a function of variance or EV in either area.

Peri-stimulus time histograms were constructed for display purposes by smoothing the cue-onset aligned spike train with a Gaussian kernel with 50 ms standard deviation *z*-scoring using the mean and standard deviation during the cue and delay epochs for each cell, and averaging across cells.

**SVM classification**. We smoothed the raw spike train using a Gaussian kernel of 50 ms standard deviation and measured the average smoothed firing rate in each trial in the interval 0–800 ms after cue onset. We then evaluated decoding accuracy for each pairwise classification (e.g., EV3 vs EV6, variance 1 vs variance 4, etc) using the data pooled across all the neurons in an array. To construct the pooled dataset, we randomly selected *m* trials from each neuron and every condition, where *m* was equal to the minimum number of trials across all neurons and all conditions. We used a fivefold cross-validation procedure with 200 repetitions to compute decoding accuracy in the original dataset and repeated the procedure after randomly shuffling trial labels to compute the baseline accuracy expected purely by chance.

**LFP pre-processing**. The raw LFP from each electrode and trial were measured from 1200 ms before to 2000 ms after cue onset, notch filtered at 60 Hz, low pass

filtered at 100 Hz, and subjected to a linear trend removal. The traces from each session were then pooled and subjected to a two-step cleaning procedure to remove outliers in, respectively, the frequency and time domains. For the first step that removed outliers in the frequency domain, we calculated the power spectrum of each LFP trace in the range of 0–90 Hz (using a multitaper method with four tapers) and characterized each trial with a five-dimensional vector containing the sum of the logarithm of the power spectrum in five frequency bands (0.5–4, 4–8, 8–12, 12–30, and 30–90 Hz). We then reduced the dimensionality of each session's dataset to two principal components using principal component analysis and clustered this two-dimensional dataset using Gaussian Mixture Models (*fitgmdist* function in the MATLAB statistics and machine learning toolbox). This procedure produced, for each session, one or two "dense" clusters that contained most of the session's data, and one or two "sparse" clusters containing the remaining trials, in which the LFP power in at least one frequency band was an outlier. We discarded the trials in the sparse clusters. In addition, we discarded trials that were identified as outliers within the dense clusters—i.e., for which the Mahalanobis distance to all other trials in the cluster was above the 90th percentile. The trials surviving the first step were subjected to a second step that removed outliers in the time domain. To this end, we computed the peak-to-peak amplitude of the broadband LFP trace in each trial, $z$ transformed these values, and removed trials for which this measure was more than half a standard deviation away from the mean across all trials. This was a conservative cleaning procedure that removed all the trials with poor signal quality due to a variety of reasons (e.g., signal-to-noise ratio, artifact, or saturation). Overall, 39.5% of trials (12.3–77.4% across sessions) were excluded after pre-processing.

**LFP power spectrum**. For each trial that was accepted for analysis, we calculated the LFP power spectrum in 1 Hz frequency bands using Morlet wavelet transformation (*ft_freqanalysis* function of the FieldTrip toolbox[58]). The power in each band was then $z$-scored relative to all the trials and time points within the session, and normalized relative to the trial's baseline using the following equation:

$$\text{relative power change}\,(t,f) = \frac{\text{power}_{tf} - \overline{\text{baseline}_f}}{\overline{\text{baseline}_f}}, \qquad (1)$$

where $\text{power}_{tf}$ is the power at time $t$ and frequency $f$, and $\overline{\text{baseline}_f}$ is the power in frequency $f$ during the 300 ms interval before cue onset on the same trial. Normalization relative to the frequency-specific baseline accounted both for trial-by-trial variability and $1/f$ power distribution[36].

**GLM of LFP power spectrum**. The *Relative Power Change* quantity from Eq. (1) produced a time-frequency map of normalized LFP power for each trial and electrode. To determine how these maps varied as a function of uncertainty and EV, for every trial we pooled the trials across the electrodes of an array, and fit this pooled dataset using a GLM with factors of [EV, variance, EV × variance, Cue location] assuming a normal distribution and identity link function. This produced a time-frequency map of coefficients measuring the effects of EV and variance, controlling for any visuo-spatial response and EV × variance interaction (Fig. 5). To identify ROI within the GLM coefficient maps, we divided the cue and delay periods into 200 ms epochs, and identified frequencies for which the coefficients for a variable were significantly different from 0 with the same sign in both monkeys (Kruskal–Wallis test with false discovery rate (FDR) correction).

**Field–field coherence**. Field–field coherence was measured using weighted phase lag index (WPLI)[37]. The WPLI uses imaginary part of the cross-spectrum to remove the volume conduction effect. Within a session, for every task condition the phase locking index was calculated across trials and LFP channel pairs for every time and frequency. GLMs with EV, variance, and EV × variance factors were then fitted to the coherence maps from different sessions, assuming normal distribution and identity link function.

**Spike-field coherence**. We used the FieldTrip toolbox[58] to calculate the power spectrum for the trial-by-trial LFP using multitaper analysis, and the *ft_spike-triggeredspectrum* function to measure the phase in frequencies of 4–47 Hz. We estimated SFC using the average Pairwise Phase Consistency index (PPC2, *ft_spiketriggeredspectrum_stat* function), which is known to minimize biases due to low spike counts and volume conduction[35,37]. For every pair of neuron-LFP channel, in each task condition and frequency, PPC2 was calculated across all spikes that the cell fired in the corresponding task condition. PPC2 values of all cell–LFP pairs (excluding pairs in which the neurons did not emit any spikes) were submitted to non-parametric analyses to detect influences of EV and variance ($n$ ranging between 136,368 and 164,880 across conditions). In the frequency plots, $p$ values were corrected for comparison across frequencies using the FDR correction method.

**Reporting summary**. Further information on experimental design is available in the Nature Research Reporting Summary linked to this paper.

## Data availability

The summary statistics are available within the article and its data supplement. All other data are available from the corresponding author upon reasonable request.

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

## Acknowledgements

The work was supported by a generous gift of surgical equipment from Synthes, Inc. and a McKnight Foundation Memory and Cognitive Disorder Award to J.G.

## Author contributions

N.C.F. and J.G. designed the experiment. N.C.F., S.A.S., M.C., M.S., R.L., and J.G. implemented the experiment and collected the data. B.T., S.K., R.L., and J.G. analyzed the data and wrote the manuscript.

## Competing interests

The authors declare that they have no competing interests. J.G. is an Editorial Board Member for *Communications Biology*, but was not involved in the editorial review of, nor the decision to publish, this article.
