## [Peer Review File · Communications Biology]

Reviewers' comments:

Reviewer #1 (Remarks to the Author):

The authors carried out an experiment in which animals were given a cue that indicated the mean and variance of the outcome associated with an eye movement. Following the eye movement, the animals were stochastically rewarded according to the distribution indicated by the cue they had been shown. While the animals carried out the task, neural activity was recorded in dlPFC and parietal area 7A. Single neurons coded either the mean or the variance of the outcome, but rarely coded both. Positive and negative correlations were about equally distributed in the population. Noise correlations were stronger between cells which had similar tuning, than cells which had different tuning. Finally, analyses of LFPs showed that there was a consistent negative encoding of EV by alpha/low beta in both areas. Furthermore, there was an asymmetric flow of information about variance, characterized by spike field coherence. Spike field coherence was stronger for high variance from 7A to dlPFC and higher for low variance from dlPFC to 7A.

The results in this paper are of general interest. The study was well-carried out, and the paper is written clearly. Overall, this is very straightforward. I have a few suggestions.

Comments

1. Perhaps a bit more analysis of the behavioral data would be useful. Were there any reaction time effects of mean or variance? Did these relate to neural activity in any way?
2. If there are RT effects, did they relate to the licking data? Did mean and variance influence RT together or through an interaction?
3. It would be useful to put a p-value on this statement - perhaps you could compare this using a binomial? "and the fraction of neurons with main effects of both factors was below 3% in each area - lower than would be expected by chance"
4. Ilya Monosov has published a few recent papers out of his own lab, looking specifically at uncertainty. It would be worth referencing these.

Reviewer #2 (Remarks to the Author):

In this study, the authors recorded neuronal activity from the dorsolateral prefrontal cortex and parietal cortex of monkeys performing in an instructed saccade task with cues that signalled the variance ('risk') and expected value (EV) of liquid rewards. Following cue presentation, the animals were instructed to make a saccade to either the left or right location; correct saccades resulted in delivery of liquid reward according to the cued reward distribution (i.e. specified by the cued EV and variance). Single neurons and population activity encoded both risk and EV, but independently of each other; both variables were encoded by separate populations of neurons, as evidenced by single-neuron betas, noise correlations, and SVM-decoding. Further, risk was associated with reduced LFP power at specific frequency bands, possibly indicating enhanced task engagement or attention.

This is an interesting, novel, well-conducted study. To my knowledge, this is the first report that examines the neuronal processing of reward uncertainty in both prefrontal and parietal cortex. The stimuli and task are well-controlled and the data analysis is sophisticated. I have a few points that should be addressed to further strengthen the paper before publication.

Major points:

1. In studies of reward uncertainty / risk, the subjective behavioral relevance of this variable to the animals is important. Most (though not all) previous studies typically reported that the monkeys were risk-seeking for small rewards. Understanding the subjective relevance of risk to the animals is important as this can influence the neuronal responses, beyond 'objective' risk. The licking data seem to suggest risk-seeking within the used reward range, although choice data are not shown. I suggest to mention that, although choices were not explicitly studied in this paper, behavioral licking data indicated that the animals were risk-seeking, consistent with previous studies.
2. In relation to the above point and Fig. 1D, it would be important to show the licking data separately for each monkey. If there are differences in the relationship between risk and licking between animals, then it would be appropriate to discuss these differences with respect to the animals' risk attitude and possible effects on neuronal coding of risk.
3. What were the error rates with which each monkey performed the task; did error rates differ for the various risk/EV levels? This is important, as differential error rates could modify the cued risk and EV levels.
4. Did the animals' reaction times reflect risk or EV? It would be helpful to do a focused GLM analysis on the reaction times. Such a relationship could support the authors' point that uncertainty processing might modulate task engagement.
5. The single-neuron examples from Fig S2 should be included in the main figures; maybe as panels in Fig. 2. It would be nice to see rasterplots for an example neuron.
6. Previous studies found neuronal encoding of decision variables in relation to specific actions, e.g. left-right saccades or arm movements: action value (striatum: Samejima et al., 2005, Science; Lau and Glimcher, 2008, DLPFC: Tsutsui et al. 2016, Nature Communications) and action risk (DLPFC: Grabenhorst et al., 2019, eLife). To link the present paper to previous studies of risk and value in decision-making context, it would be helpful to know whether neurons in DLPFC and parietal cortex in the present study also coded value and risk specifically when these variables were presented on the left or right side. Due to the task design, the authors can address this question very nicely: although it is important to note that this is not a decision task, the randomized presentation of the EV/risk cue on the left or right could uncover such spatially specific risk and EV coding. Testing for this effect would require an additional GLM that codes risk and EV separately for the left and right side presentations.

Minor points:

7. Line 48 and 229: A previous study (reference 14; Grabenhorst et al., 2019, eLife) had shown that neurons in DLPFC code risk independently of value and vice versa.

8. Line 61: It is unclear what the authors mean by "decision incentives". A conditioned stimulus would have predictive reward value and uncertainty regardless of whether it is presented in a choice or non-choice situation. The choice situation would involve additional neuronal processing, including value comparison and choice signalling; however, the meaning of a presented cue in terms of the animal's reward outcome would not differ per se.

9. Neuronal recordings: what counted as a 'neuron', i.e. did the authors consider the possibility that the same neuron was recorded on different days?

Reviewer #3 (Remarks to the Author):

The authors investigated the monkey fronto-parietal network by using two multi-channel electrode arrays, and obtained the following results. Single-neuron activity of area 7A and aIPFC separately encode variance and EV. Neurons that code the same factor are more associated than neurons with mixed selectivity. Responses of neuronal populations effectively discriminate between levels of variance and EV. Oscillatory LFP are affected by variance and EV in two frequency bands. The SFC suggests that higher uncertainty enhances information transmission from 7A to dIPFC and reduces from dIPFC to 7A.

This research is technically sound and presents new insights into effects of reward uncertainty on information transmission between the fronto-parietal network. It will be interesting to researchers of higher cognitive functions, such as decision-making under uncertainty.

A concern I found is the task design. The authors considered that the distance between the two bars in the visual cues signaled variance of reward probability and that neural responses to the cues encoded the variance. However, another possibility is that those neurons might respond only to the visual cues consisting of the two bars irrespective of cue position, not to the variance of the upcoming probabilistic rewards. If the authors agree with this possibility, a brief remark concerning this possibility is desirable in the discussion.

The legend of Fig. 6B does not include explanation of what the black and gray curves indicate. Although I suspect that the black and gray curves correspond to 7A and dIPFC, respectively, as in Fig. 5, explicit description will improve readability.

In the GLM of LFP power spectrum paragraph of Methods, "Fig. 3" (line 388) may be an error. Is it Fig. 4?

A recent work (Nakamura and Komatsu, *Brain Research*, 1707 (2019) 79-89) shows that neural activity of dIPFC codes reduction in probabilistic uncertainty (i.e., information value), and might be a reference that supports the idea that the brain uses uncertainty to engage the prefrontal cortex in uncertain conditions.

Jun 16, 2020

=====

Referee expertise:

Referee #1: decision making, primate recording, pre-frontal cortex

Referee #2: Decision-making, reward processing, electrophysiology, computational modeling

Referee #3: computational neuroscience, reward processing

Reviewers' comments:

Reviewer #1 (Remarks to the Author):

The authors carried out an experiment in which animals were given a cue that indicated the mean and variance of the outcome associated with an eye movement. Following the eye movement, the animals were stochastically rewarded according to the distribution indicated by the cue they had been shown. While the animals carried out the task, neural activity was recorded in dIPFC and parietal area 7A. Single neurons coded either the mean or the variance of the outcome, but rarely coded both. Positive and negative correlations were about equally distributed in the population. Noise correlations were stronger between cells which had similar tuning, than cells which had different tuning. Finally, analyses of LFPs showed that there was a consistent negative encoding of EV by alpha/low beta in both areas. Furthermore, there was an asymmetric flow of information about variance, characterized by spike field coherence. Spike field coherence was stronger for high variance from 7A to dIPFC and higher for low variance from dIPFC to 7A.

The results in this paper are of general interest. The study was well-carried out, and the paper is written clearly. Overall, this is very straightforward. I have a few suggestions.

We thank the reviewer for the supportive remarks and constructive suggestions.

Comments

1. Perhaps a bit more analysis of the behavioral data would be useful. Were there any reaction time effects of mean or variance? Did these relate to neural activity in any way? We thank the reviewer for bringing up this point. Although we had extensively analyzed the monkeys' saccades, we found that they were not consistently affected by variance or EV or correlated with the licking effects. We had stated this very briefly in the original manuscript, and expand on this point in the revised **Results** (p. 5,

2nd paragraph describing **Fig. 1d**). We also found no correlation between the behavioral effects and the encoding of variance and EV by individual cells (p 6, 2nd paragraph) or LFP (p. 9, top paragraph).

2. If there are RT effects, did they relate to the licking data? Did mean and variance influence RT together or through an interaction? We found no effects on RT, as noted above. We also found no relation between the effects of EV and variance on licking and those on RT, and no interaction between the effects of EV and variance on licking and cue location - consistent with previous reports that licking and saccades react differently to reward manipulations. We now report the results in the same section (p. 5, 2nd paragraph the text describing **Fig. 1d**).

3. It would be useful to put a p-value on this statement - perhaps you could compare this using a binomial? “and the fraction of neurons with main effects of both factors was below 3% in each area - lower than would be expected by chance” In evaluating the percentage of cells, we adopted the standard approach of comparing the percentage of significant cells to our criterion of $p = 0.05$. We reasoned that a percentage of 3% is below this standard criterion, indicating that it is likely to have occurred by chance. However, we acknowledge that, although this reasoning is valid when discussing individual effects, it is more complicated when discussing a conjunction of effects (which may be the basis of the reviewer’s question). We therefore removed the statement the reviewer mentioned, and base our case on the other clear evidence supporting this statement – the lack of correlation between the GLM coefficients, and the analysis of subsets of cells that has been moved to new **Fig. 3**.

4. Ilya Monosov has published a few recent papers out of his own lab, looking specifically at uncertainty. It would be worth referencing these. We had included several citations of the Monosov work. We now added a few more recent citations from Ilya’s own lab, including a recent study that records from the ACC (**Introduction** and **Discussion**).

Reviewer #2 (Remarks to the Author):

In this study, the authors recorded neuronal activity from the dorsolateral prefrontal cortex and parietal cortex of monkeys performing in an instructed saccade task with cues that signalled the variance ('risk') and expected value (EV) of liquid rewards. Following cue presentation, the animals were instructed to make a saccade to either the left or right location; correct saccades resulted in delivery of liquid reward according to the cued reward distribution (i.e. specified by the cued EV and variance). Single neurons and population activity encoded both risk and EV, but independently of each other; both variables were encoded by separate populations of neurons, as evidenced by single-neuron betas, noise correlations, and SVM-decoding. Further, risk was associated with reduced LFP power at specific frequency bands, possibly indicating enhanced task engagement or attention.

This is an interesting, novel, well-conducted study. To my knowledge, this is the first report that examines the neuronal processing of reward uncertainty in both prefrontal and parietal cortex. The stimuli and task are well-controlled and the data analysis is

sophisticated. I have a few points that should be addressed to further strengthen the paper before publication.

We thank the reviewer for the supportive remarks and constructive suggestions.

Major points:

1. In studies of reward uncertainty / risk, the subjective behavioral relevance of this variable to the animals is important. Most (though not all) previous studies typically reported that the monkeys were risk-seeking for small rewards. Understanding the subjective relevance of risk to the animals is important as this can influence the neuronal responses, beyond 'objective' risk. The licking data seem to suggest risk-seeking within the used reward range, although choice data are not shown. I suggest to mention that, although choices were not explicitly studied in this paper, behavioral licking data indicated that the animals were risk-seeking, consistent with previous studies. This is an excellent point, which we also wondered about. We examined this question by testing the monkeys, after the recordings were complete, on a choice version of the task. The choice test had to be done after the neural recordings to avoid contaminating the neural data with potential choice-specific strategies. Thus, these data do not provide detailed neural-behavioral correlations. However, they reveal intriguing correlations between the overall features of the neural responses and individual risk attitudes, which we now present the results in an additional section at the end of the **Results** (p. 10/11) and in a new supplementary figure (**Fig. S6**).

As the reviewer can see, the monkeys had distinct risk attitudes, with monkey 1 being risk seeking and monkey 2 risk averse. This, in turn, correlated with the monkeys' relative sensitivity to variance versus EV. The risk averse monkey was more sensitive to variance relative to EV, while the risk seeking monkey was more sensitive to EV relative to variance. Differences in relative sensitivity were found in licking, RT and, at the neural level, in LFP power and fraction of selective cells (but not in SFC).

We agree that the results are intriguing. However, because we only tested 2 monkeys and did not compare neural responses in choice and non-choice contexts, we obviously cannot draw strong conclusions and initially chose not to present those data. We now present the results as an interesting point for further research – and are happy to include or remove it at the reviewers' and editors' discretion.

2. In relation to the above point and Fig. 1D, it would be important to show the licking data separately for each monkey. If there are differences in the relationship between risk and licking between animals, then it would be appropriate to discuss these differences with respect to the animals' risk attitude and possible effects on neuronal coding of risk. **Fig. S6** and the related discussion present individual data, as described above.

3. What were the error rates with which each monkey performed the task; did error rates differ for the various risk/EV levels? This is important, as differential

error rates could modify the cued risk and EV levels. This is a good point, which we address in the Results (p. 5, 1st paragraph, description of **Fig. 1**). There were no effects of variance or EV on error rates.

4. Did the animals' reaction times reflect risk or EV? It would be helpful to do a focused GLM analysis on the reaction times. Such a relationship could support the authors' point that uncertainty processing might modulate task engagement. This is a good point that is also mentioned by Reviewer 1. As we state in our response to Reviewer 1 (points 1 and 2), we had extensively analyzed the monkeys' saccades, but found no consistent effects of variance or EV, no interaction between the encoding of EV/variance and of the cue location (**Results**, p. 5, 2nd paragraph, the text describing **Fig. 1d**) and no correlation between the behavioral effects and the encoding of variance and EV by individual cells (**Results**, p. 6, 2nd paragraph) or LFP (**Results**, p. 9, top paragraph). Although RT effects may have indicated arousal, as the reviewer suggests, we believe the task we used may have been too simple to elicit them.

5. The single-neuron examples from Fig S2 should be included in the main figures; maybe as panels in Fig. 2. It would be nice to see rasterplots for an example neuron. We thank the reviewer for this suggestion and have now moved **Fig. S2** to the main text (new **Fig. 3**). Although we tried to accommodate raster plots, we ultimately felt that including these plots unnecessarily complicates the exposition without bringing much more information than the peri-stimulus time histograms we show. Thus, we opted to leave out the raster plots but we would be happy to provide them to the reviewer if he/she wishes.

6. Previous studies found neuronal encoding of decision variables in relation to specific actions, e.g. left-right saccades or arm movements: action value (striatum: Samejima et al., 2005, Science; Lau and Glimcher, 2008, DLPFC: Tsutsui et al. 2016, Nature Communications) and action risk (DLPFC: Grabenhorst et al., 2019, eLife). To link the present paper to previous studies of risk and value in decision-making context, it would be helpful to know whether neurons in DLPFC and parietal cortex in the present study also coded value and risk specifically when these variables were presented on the left or right side. Due to the task design, the authors can address this question very nicely: although it is important to note that this is not a decision task, the randomized presentation of the EV/risk cue on the left or right could uncover such spatially specific risk and EV coding. Testing for this effect would require an additional GLM that codes risk and EV separately for the left and right side presentations. We agree that the interaction with spatial selectivity is important, and our findings clearly show that the EV/variance sensitivity was independent of location selectivity. In addition to the analyses noted above, we added a new paragraph clearly outlining the evidence for this with reference to **Fig. S2** (**Results**, p. 6, penultimate paragraph). We also added a discussion note regarding the difference between our finding and previous reports of spatially specific modulations (**Discussion**, p. 12, 1st paragraph).

Minor points:

7. Line 48 and 229: A previous study (reference 14; Grabenhorst et al., 2019, eLife) had shown that neurons in DLPFC code risk independently of value and vice versa. This is a good point and we added a paragraph discussing the results of Grabenhorst and similar papers (**Discussion**, p. 12). This discussion includes the fact that they are showing spatial rather than non-spatial effects and speculates that the reason for our different findings may be in the type of task we used (expectation versus decision based).

8. Line 61: It is unclear what the authors mean by "decision incentives". A conditioned stimulus would have predictive reward value and uncertainty regardless of whether it is presented in a choice or non-choice situation. The choice situation would involve additional neuronal processing, including value comparison and choice signalling; however, the meaning of a presented cue in terms of the animal's reward outcome would not differ per se. We simply meant that the monkeys could not choose in our task (i.e., did not make an incentivized choice). We clarified the language in the revision.

9. Neuronal recordings: what counted as a 'neuron', i.e. did the authors consider the possibility that the same neuron was recorded on different days? This is a good point. As the reviewer may know, a full quantitative answer is complex (would almost require a separate methods paper). For this study, we followed the state of the art in the field of visually verifying that the waveforms changed across days; we added a note on this in the **Methods** (p. 13, penultimate paragraph).

Reviewer #3 (Remarks to the Author):

The authors investigated the monkey fronto-parietal network by using two multi-channel electrode arrays, and obtained the following results. Single-neuron activity of area 7A and aIPFC separately encode variance and EV. Neurons that code the same factor are more associated than neurons with mixed selectivity. Responses of neuronal populations effectively discriminate between levels of variance and EV. Oscillatory LFP are affected by variance and EV in two frequency bands. The SFC suggests that higher uncertainty enhances information transmission from 7A to dIPFC and reduces from dIPFC to 7A.

This research is technically sound and presents new insights into effects of reward uncertainty on information transmission between the fronto-parietal network. It will be interesting to researchers of higher cognitive functions, such as decision-making under uncertainty.

We thank the reviewer for the supportive comments and constructive suggestions.

1. A concern I found is the task design. The authors considered that the distance between the two bars in the visual cues signaled variance of reward probability and that neural responses to the cues encoded the variance. However, another possibility is that those neurons might respond only to the visual cues consisting of the two bars irrespective of cue position, not to the variance of the upcoming probabilistic rewards. If the authors agree with this possibility, a brief remark concerning this possibility is desirable in the discussion. This is a good point that was also raised by the other reviewers. As we noted above, we conclusively show that the EV/variance sensitivity was independent of location selectivity. We provide more details on this point, including that the variance and EV coefficients were equivalent for both cue locations (Results, p. 6, description of Fig. 3), and more detailed analyses of the independence of the two signals in Fig. S2; see also reply to Reviewer 1). We also state the equivalent result for LFPs (Results, p. 9, top paragraph).

2. The legend of Fig. 6B does not include explanation of what the black and gray curves indicate. Although I suspect that the black and gray curves correspond to 7A and dlPFC, respectively, as in Fig. 5, explicit description will improve readability. The colors indicate low and high variance (gray: var = 0; black: var = 4). The answer was in the legend and, to make it more salient, we now also note it in the title to Fig. 7B (former 6B).

3. In the GLM of LFP power spectrum paragraph of Methods, “Fig. 3” (line 388) may be an error. Is it Fig. 4? Thank you for detecting this error. We now corrected it to refer to Fig. 5.

4. A recent work (Nakamura and Komatsu, Brain Research, 1707 (2019) 79-89) shows that neural activity of dlPFC codes reduction in probabilistic uncertainty (i.e., information value), and might be a reference that supports the idea that the brain uses uncertainty to engage the prefrontal cortex in uncertain conditions. This is a good point, we now included the reference to Nakamura and Komatsu in the Introduction and Discussion. We note that the study focused on responses to the value of information (VOI) which by definition, combine rather than dissociating value and uncertainty. Thus, while we mention this relevant study, we do not discuss it in detail as we felt this would take us too far off track for this particular paper.

REVIEWERS' COMMENTS:

Reviewer #1 (Remarks to the Author):

I have no further comments. The authors have addressed my concerns.

Reviewer #2 (Remarks to the Author):

The authors have done a very thorough revision and convincingly addressed all points that were raised.